# Cystitis Induces Altered CREB Expression Related with Micturition Reflex

**DOI:** 10.3390/medicina58091210

**Published:** 2022-09-02

**Authors:** Taesoo Choi, Dong-Gi Lee

**Affiliations:** Department of Urology, School of Medicine, Kyung Hee University, Seoul 05278, Korea

**Keywords:** urinary bladder, *Escherichia coli*, cystitis, ganglia, spinal, rats

## Abstract

*Background and objectives:* Bladder stimulation upregulates neurotrophins associated with voiding reflex. Bacterial cystitis can be a stimulant that activates this system, resulting in a pathological state. Phosphorylated responsive element of binding protein (p-CREB) is a pivotal transcriptional factor in the neurotrophin signaling cascade. The goal of our study was to examine the change in expression of p-CREB in dorsal root ganglia (DRG) of rats after uropathogenic *Escherichia coli* infection of the bladder. *Materials and methods:* A total of 19 adult female Sprague–Dawley rats were induced with acute *E. coli* infection (*n* = 7), chronic *E. coli* infection (*n* = 6), or served as controls (*n* = 6). In each group, the profiles of p-CREB cell were counted in 6–10 sections of each of the DRG collected. DRG cells exhibiting intense nuclear staining were considered to be positive for p-CREB immunoreactivity (p-CREB-IR). *Results:* Overall, the immunoreactivity of p-CREB was examined in smaller cell profiles with nuclear staining or nuclear and cytoplasmic staining in the DRGs (L1–L6, S1). In the chronic cystitis group, p-CREB-IR in the L1–L6 and S1 DRG was significantly higher than the control group (*p* < 0.05). Further, p-CREB-IR in the L3–L6 and S1 DRG of the chronic cystitis group was significantly greater than that in the acute cystitis group (*p* < 0.05). In the control and acute cystitis groups, p-CREB-IR in the L4–L5 DRG was significantly lower than that found in the other DRG sections (*p* < 0.05). *Conclusions:* Altogether, acute or chronic *E.coli* cystitis changed the immunoreactivity of p-CREB in lumbosacral DRG cells. In particular, chronic *E. coli* infection triggered p-CREB overexpression in L1–L6 and S1 DRG, indicating subsequent pathologic changes.

## 1. Introduction

Ordinary micturition requires coordinated contraction between the detrusor muscle and striated muscle of the external urethral sphincter that is controlled by spinal and supraspinal circuitry. Any unnatural alteration of this signaling system, called micturition reflex, may lead to continued voiding problem in patients. In clinical practice, patients with bacterial cystitis often exhibit prolonged lower urinary tract symptoms, such as frequency, urgency, tenesmus, painful urination, or suprapubic pain, even after complete treatment.

Bladder stimulation upregulates neurotrophins, which contribute to voiding reflex. Increased circulating neurotrophins in urine and urothelium have been demonstrated in patients with interstitial cystitis/bladder pain syndrome [1]. Increased urinary nerve growth factor (NGF), one of the neurotrophins, was ALSO reported in patients with overactive bladder, benign prostate hypertrophy, or stress urinary incontinence in a similar vein [2,3]. Neurotrophin-mediated functional changes in bladder and alteration of somatic sensitivity may lead to the altered expression of comprehensive neurotrophin/receptor systems, resulting in irreversible changes in the lower urinary system. Bacterial cystitis can also be a stimulant that activates this system, resulting in a pathological state.

The cyclic adenosine monophosphate response element binding protein (CREB) is a representative transcription factor involved in the regulation of different genes. The phosphorylation of CREB has been demonstrated to be required for CREB-mediated transcription. Previously, an animal study revealed that phosphorylated CREB (p-CREB) in bladder afferent neurons was upregulated in the dorsal root ganglia (DRG) in the lumbosacral region after chemical cystitis, which induced subsequent alterations in the diverse characteristics of bladder afferent neurons in DRGs that influence the central reflex micturition pathways [4]. These changes suggest remarkable reorganization of reflex connections between bladder and spinal cord, which affects micturition reflexes eventually.

In vitro preceding researchers demonstrated that the activation of downstream signaling molecules, specific transcription factors such as CREB, are crucial steps in neurotrophin signaling cascades [5,6]. These signals induce related transcriptions and cause their expression and long-term changes in cells by: mediating neurotransmitter phenotype; influencing synaptic reorganization; inducing cell survival or differentiation; and controlling and function in a target organ.

We hypothesized that the changes in micturition patterns after acute or chronic bacterial cystitis involves an alteration of the urinary tract, especially bladder physiology. In the present study, we used a specific antibody to measure the phosphorylation of CREB in neurons within the spinal cord and DRG, to ultimately evaluate the altered expression of the transcription factor, p-CREB, in DRG under control or infection conditions using one of the most common uropathogens, *Escherichia coli* (*E. coli*), in rat urinary bladder. There are few studies on the relationship between p-CREB expression and cystitis, and to the best of our knowledge this study is the first report on the p-CREB expression after bacterial cystitis.

## 2. Materials and Methods

### 2.1. Experimental Animals

Adult female Sprague–Dawley rats weighing 280 ± 20 g were employed in the study. All rats were verified to initially have a negative urine culture. To maximize the efficiency in this study, fine sections of the DRG were harvested equally and then divided into three groups sequentially to meet our experimental condition. For each set of experiments, six control, seven experimental animals with acute cystitis, and six experimental animals with chronic cystitis were used. Whole environmental control before the experiment was equivalent for the study groups. The room temperature was maintained at 22 ± 2 °C, with 40–70% relative humidity, and a 12 h light and dark cycle. Because CREB phosphorylation is quite sensitive to sensory stimuli, uncontrolled variables were minimized. All research experiments were conducted legally and followed the protocols approved by the Animal Care Committee of the Animal Center at Kyung Hee University; they also complied with the guidelines from the Korean National Health Institute of Health Animal Facility. All efforts were intended to prevent any potential animal abuse, such as pain, stress, or distress.

### 2.2. Induction of E. coli Cystitis

*E. coli* cells, fimH+, sfa+, papA+, were cultivated on MacConkey agar for 24 h at 37 °C. These colonies were mixed with phosphate-buffered saline (PBS), then suspended to a concentration of 2 × 108 bacteria/mL using photospectrometry [7]. A sterile catheter (outer diameter, 0.50 mm) was indwelled into the bladder through the urethra under isoflurane anesthesia (2% in oxygen). After emptying the urine, 0.5 mL of the inoculum was instilled into the bladder via the catheter, resulting in a dose of 1 × 10^8^
*E. coli* per rat. Thereafter, the urethral meatus was tied with black silk for 4 h to keep the inoculum in the bladder.

### 2.3. Microscopic Examination

The bladder specimen was embedded in the paraffin and fine 4 μm sections were created for hematoxylin and eosin staining. The pathologic changes in the bladder mucosa and submucosa were evaluated under an optical microscope.

### 2.4. Study Groups

Rat models with acute and chronic *E. coli* cystitis were induced and examined based on previous studies (Figure 1). For chronic *E. coli* cystitis, rats received bladder instillation once per week for one month. Rats were exsanguinated after one week to obtain DRGs. For acute *E. coli* cystitis, rats received a single bladder instillation of *E. coli* as described above. Control rats received volume-matched (0.5 mL) sterile saline instillation. The rats with acute infection or control rats were killed the next day to obtain DRGs.

### 2.5. Perfusion and Tissue Harvesting

After control or *E. coli* infection, all rats were anesthetized with sodium pentobarbital (50 mg/kg, intraperitoneal), then perfused with 0.05 M phosphate-buffered saline (PBS), followed by 4% paraformaldehyde. After perfusion, the dorsal L1–S1 vertebrae were removed based on the previously described representation of urinary bladder circuity [8,9]. Thereafter, the DRGs in the spinal cord were promptly harvested and postfixed for 6 h. Spinal cord segments were classified based on two major criteria [10]: (1) T13 DRG located just below the last rib; and (2) L6 vertebra as the last mobile vertebra, followed by the sacral vertebrae. The L6 DRG was recognized as the smallest ganglia while the L5 DRG was recognized as the largest. DRG from the L1 to S1 spinal cord segments were sectioned at 20 μm each on a freezing microtome. These DRGs were specifically selected for analysis associated with an autonomic reflex at the spinal cord level. Specimens from the control group with an uninfected spinal cord were handled in a same manner to that described above.

### 2.6. Immunohistochemistry for p-CREB

DRG sections from the control and acute/chronic cystitis groups were handled for p-CREB-immunoreactivity (p-CREB-IR) utilizing an on-slide processing technique [11]. To minimize the risk of variation in staining and background, all groups were processed simultaneously between tissues and between animals. DRG sections were incubated overnight at constant temperature with anti–p-CREB antibody (1:1000; Cell Signaling Technology, Danvers, MA, USA) in 1% goat serum and 0.1 M KPBS (Phosphate Buffer Solution with potassium), then washed three times with 0.1 M KPBS. DRG sections were then incubated with Cy3-conjugated IgG (1:500; Jackson Immunoresearch, West Grove, PA, USA) for 2 h at room temperature. Several rinses with 0.1 M KPBS were performed, followed by mounting with Citifluor (Citifluor, London, UK) on slides before placement of the coverslip [4,12].

### 2.7. Assessment of IR-Stained DRG Cells

Specimen with specific staining was compared with that of matched negative controls. As shown in Figure 2, DRG cells exhibiting intense nuclear staining were determined positive to measure the IR of p-CREB. Cells with intense nuclear staining have randomly exhibited cytoplasmic staining more than the background level observed in matched controls. DRG cells expressing p-CREB staining but without certain nuclear staining were not determined positive in this study. There was no further division into categories of staining levels depending on the characteristics of positive cells.

### 2.8. Data Analysis

Tissues with immunohistochemical staining were examined under a fluorescence photo-microscope at 100–400× magnification. In the DRG from acute/chronic cystitis rats and control, p-CREB cell profiles were measured in 6–10 sections of each DRG (L1–S1). Only profiles with clear nuclear staining were counted. The cell profiles of p-CREB IR in each DRG section are presented as mean ± standard deviations (SD). Cell diameters of DRG cells were measured along the long and short axes. The average of them was presented as the mean diameter (μm). Comparisons between the control and cystitis groups were carried out using ANOVA. The collected data were transformed to meet the requirements of ANOVA and analyzed using SPSS 28.0 software. A *p*-value < 0.05 was considered statistically significant.

## 3. Results

The expression of p-CREB in L1–S1 DRG was analyzed after single or four rounds of *E. coli* instillation in the bladder. Typically, CREB-immunoreactive staining was localized to the nuclei.

### 3.1. Quantitative Analysis of p-CREB in Each Segment of the DRG

The p-CREB-IR was measured in nuclear profiles and primarily restricted to smaller diameter cells in all segments of the DRG (L1–S1). The extent of expression of p-CREB in each group was calculated in accordance with the DRG segment. In the acute cystitis group, the cell profiles of p-CREB-IR were 31.3 ± 4.42 (L1), 32 ± 8.85 (L2), 28.79 ± 5.86 (L3), 20.17 ± 6.05 (L4), 25.64 ± 7.26 (L5), 29.14 ± 6.59 (L6), and 36 ± 8.49 (S1). In the chronic cystitis group, the cell profiles of p-CREB-IR were 33.9 ± 1.40 (L1), 36.82 ± 1.49 (L2), 38.15 ± 1.69 (L3), 31.79 ± 2.07 (L4), 33.25 ± 2.17 (L5), 41.94 ± 1.94 (L6), and 42.57 ± 2.99 (S1). Finally, in the control, the cell profiles of p-CREB-IR were 28.69 ± 8.85 (mean ± SD; L1), 29.44 ± 8.41 (L2), 28.33 ± 7.83 (L3), 22.81 ± 5.04 (L4), 23.85 ± 7.19 (L5), 32.22 ± 7.33 (L6), and 31.4 ± 9.20 (S1), as described in Table 1.

### 3.2. Time Series Analysis of p-CREB IR in DRG with E. coli Cystitis

The sequential analysis of p-CREB expression in spinal cord neurons after bacterial cystitis was assessed. Rats (*n* = 7) were examined 24 h after acute infection while other rats (*n* = 6) were examined 1 week after chronic infection. The number of p-CREB-IR cells in the acute *E. coli* cystitis group was not different from control group (*p* > 0.05). However, repeated infection induced a robust change in CREB phosphorylation in DRGs. The number of p-CREB-IR cells in the chronic *E. coli* cystitis group was significantly higher than the control group (*p* < 0.05). In the chronic *E. coli* cystitis group, the number of p-CREB-IR cells in the L3–S1 DRG was significantly higher than in the acute *E. coli* cystitis group (*p* < 0.05).

### 3.3. Changes in p-CREB IR in Bladder Afferent Cells with or without E. coli Cystitis

In the control and acute cystitis groups, as shown in Figure 3, the number of p-CREB-IR cells in the L4–L5 DRG was significantly lower than that in other segments of the DRG (*p* < 0.05). In the chronic cystitis group, the number of p-CREB-IR cells in the L1–L6 and S1 DRG was significantly higher than the control group (*p* < 0.05). Further, in the chronic cystitis group, the number of p-CREB-IR cells in the L6 and S1 DRG was significantly higher than that in the L4–L5 DRG (*p* < 0.05). Overall, the greatest changes were observed in the S1 DRG after acute and chronic cystitis.

## 4. Discussion

The goal of our study was to verify whether acute or chronic bacterial cystitis involved changes in the transcription factor, p-CREB, in the lumbosacral DRG. Remarkable increases were found in the expression of p-CREB in the DRG after bacterial infection. The putative transcription factor in the tyrosine kinase (Trk)-mediated signaling cascade, p-CREB, was induced by different types of *E. coli* exposure (single or repeated), and the magnitude of their changes differed to a certain degree. Furthermore, *E. coli* infection induced the phosphorylation of CREB in lumbosacral DRG with the course of time. The number of p-CREB-IR cells increased in all DRGs (L1 to S1) after chronic cystitis compared with those in control rats. There was no statistically significant difference in the p-CREB-IR cells between the acute infection group and the control group. We have shown the increased p-CREB expression in bladder afferent neurons after chronic cystitis. This could be explained by various factors, including changes in detrusor activity, its afferent activity, and expression in associated neurotrophic factors. The difference in basal activity in the target organ may result in altered p-CREB expression without inflammatory disease.

Previous studies have revealed the involvement of transcription factors as an important step in the Trk-mediated signaling pathway [5]. Neurotrophin/Trk signaling may play a crucial role in exhibiting p-CREB expression in bladder afferent cells after bacterial cystitis. p-CREB is presumed to be a downstream transcription factor in Trk-related signaling pathways in micturition reflex after recurrent *E. coli* cystitis. Similarly, bladder afferent cells tend to respond to nerve growth factor and brain-derived neurotrophic factor either from the spinal cord or urinary bladder [13]. Neurotrophic factors can be synthesized by diverse inflammatory cells, for example lymphocytes and mast cells, causing hypersensitivity and pain with neurogenic inflammation, such as interstitial cystitis [14]. Neural interactions with target organ mediated by an increase in neurotrophins to the neuronal cell bodies in the afferent pathways might contribute to the emergence of new-onset lower urinary tract symptoms [15].

Bacterial cystitis in animal models is a well-established and is common in many research investigations into the pathophysiology of bladder dysfunction. Based on alterations in neurotrophic factors in the cystitis model, some previous studies have revealed changes in the cytokine transcripts and proteins [16,17]. Both neurotrophins and cytokines may also arouse changes in p-CREB after bacterial infection. Inflammation mediates changes in Trk expression and its phosphorylation, then induces the activation of the Trk signaling pathway in DRG [3]. The over-expressed neurotrophic factor mechanisms may increase p-CREB in bladder afferent neurons, and p-CREB-IR cells co-expressed p-Trk in rats with chronic cystitis, providing strong evidence for neurotrophin-mediated signal transduction in chronic cystitis [18]. Thus, bladder afferent neurons have two possible sources of neurotrophins after chronic bacterial cystitis: central terminals (spinal cord), and peripheral terminals (urinary bladder).

In our study, we examined the change in the downstream transcription factor, p-CREB, in DRG after *E. coli* cystitis. CREB is activated by phosphorylation. Therefore, antibodies specific to phosphorylated CREB were experimented with for this study. Previous studies and our results demonstrated little p-CREB-IR expression in DRG cells from the control group, whereas significant increase in its expression was observed in some pathologic conditions, such as renal artery occlusion or cyclophosphamide induced cystitis [4,19]. The p-CREB-IR showed upregulation in the L1–S1 DRG after chronic bacterial cystitis. Such findings oppose that of previous studies, which revealed an insignificant distribution of bladder afferents within L4–L5 DRGs, which contain only somatic afferents. Further, neurons involved in bladder function are not observed in the L4–L5 segments [9]. However, no significant changes were detected after acute infection. Our study indicated increased p-CREB-IR in bladder afferent neurons. In fact, the regulation of p-CREB triggered by neurotrophins or cytokines in DRG under pathological conditions demonstrated various patterns. Owing to this possibility, addressing the involvement of p-CREB in peripheral nerve injury, somatic and visceral inflammation and sympathetic neuron survival is crucial [20,21,22,23,24,25]. Altogether, a varied response of CREB to the inflammation of the urinary tract is suggested. Similarly, trabeculated bladder resulting from bladder outlet obstruction exhibited an increased nerve growth factor, the representative neurotrophic factors that cause neuronal hypertrophy [26]. Increased access to bladder neurotrophins may increase Trk related p-CREB expression in the bladder afferent neurons [4]. The change in neurotrophic factor expression in the bladder from different triggers, including bacterial infection, may contribute to the expression of p-CREB in DRG. Based on our results, CREB phosphorylation might contribute to lower urinary tract plasticity induced by chronic bladder infection. There are several signaling pathways and various transcription factors involved in Trk mediated signaling cascades. Among them, CREB phosphorylation may play a pivotal role in lower urinary tract characteristics. Different types of bladder afferent cells may also utilize each transcription factor in response to a certain neurotrophin exposure after bacteria-induced inflammation. It is suggested that other transcription factors could also be induced by bacterial cystitis.

There are a few limitations. We could not extend our research to reaffirm consistent results on other neurotrophins or transcription factors of Trk signaling pathways, which are expected to be closely connected to p-CREB. A few similarly themed studies on CREB activity in cystitis have been reported [27,28], but there is still a dearth of research to support the significance of p-CREB. Furthermore, our results might have been underpowered due to the relatively small sample size. To overcome this limitations, further large scale, prospective and multi-center trials could verify the precise role of p-CREB alteration that may trigger long-term pathophysiologic changes in micturition reflex.

Based on our results, acute or chronic bacterial cystitis may cause unfavorable changes in micturition, and the novel medicines modulating p-CREB expression would be one solution to care for patients with a history of urinary tract infection who are complaining of persistent urinary symptoms.

## 5. Conclusions

In summary, this study sought to elucidate changes in the IR of the transcription factor (p-CREB) in lumbosacral DRG cells after acute (24 h) or chronic (once per week for 4 weeks) *E. coli* cystitis. During chronic infection, an increased number of p-CREB-IR DRG cells was observed. Our findings indicate that bacterial cystitis, especially chronic cystitis, affects p-CREB in bladder afferent cells in the DRG, which implies the involvement of bladder afferents in the propensity of lower urinary tract plasticity after cystitis. The p-CREB expression under repetitive infection in DRG cells may be associated with sequential changes in bladder originated factors that influence the micturition reflex pathways.

## Figures and Tables

**Figure 1 medicina-58-01210-f001:**
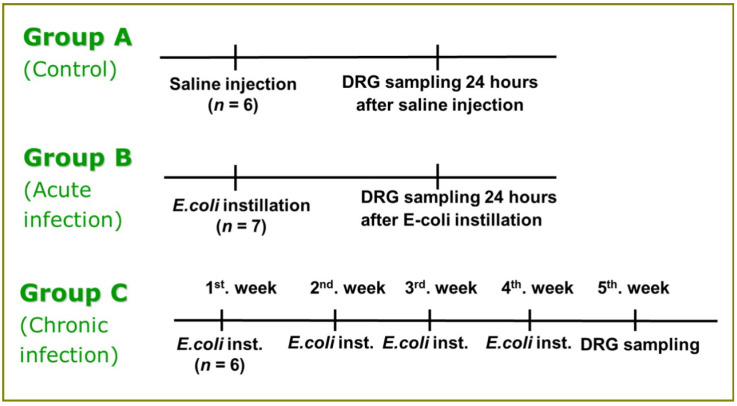
Study design of *E.coli* instillation and DRG sampling by group. A total of 19 adult female Sprague–Dawley rats were induced with acute *E. coli* infection (group B), chronic *E. coli* infection (group C), or served as controls (group A). In each group, the profiles of phosphorylated CREB cell were counted in 6–10 sections of each of the DRG collected. * DRG = dorsal root ganglia.

**Figure 2 medicina-58-01210-f002:**
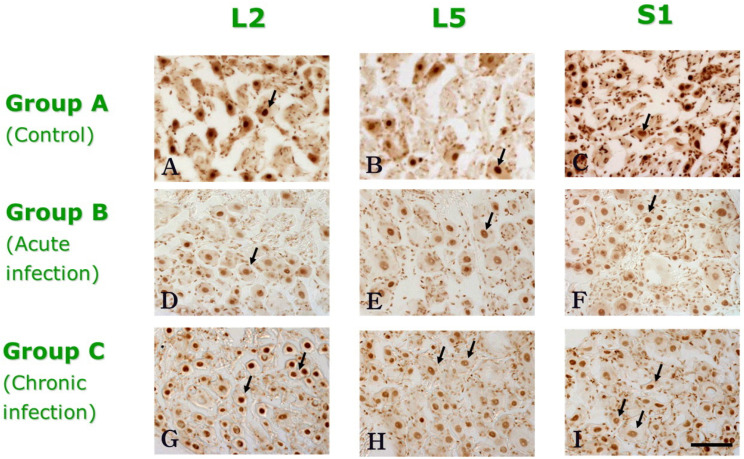
The expression of p-CREB in L2, L5 and S1 DRG in each group. In all groups, p-CREB immunoreactivity was observed in relatively small-diameter cell profiles with nuclear staining or nuclear and cytoplasmic staining in DRGs. The black arrows demonstrate the positive p-CREB immunoreactivity; DRG cells exhibiting intense nuclear staining. * Scale bar is 80 μm.

**Figure 3 medicina-58-01210-f003:**
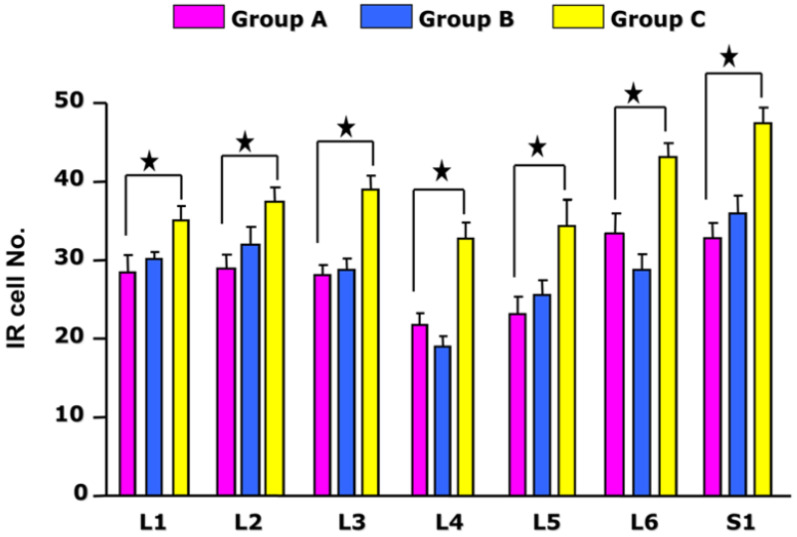
Histogram demonstrating the number of p-CREB immunoreactive cells in each group: control (group A), acute *E.coli* infection (group B) and chronic *E.coli* infection (group C). In the chronic cystitis group, p-CREB immunoreactivity (p-CREB-IR) in the L1–L6 and S1 DRG was significantly higher than control group (*p* < 0.05). Further, p-CREB-IR in the L3–L6 and S1 DRG of the chronic cystitis group was significantly greater than that in the acute cystitis group (*p* < 0.05). In the control and acute cystitis groups, p-CREB-IR in the L4–L5 DRG was significantly lower than that found in the other DRG sections (*p* < 0.05). ★ means statistical significance.

**Table 1 medicina-58-01210-t001:** Bladder afferent cells expressing p-CREB immunoreactivity.

DRG	Control (Mean ± SD)	Acute Cystitis (Mean ± SD)	Chronic Cystitis (Mean ± SD)
L1	28.69 ± 8.85	31.3 ± 4.42	33.9 ± 1.40
L2	29.44 ± 8.41	32 ± 8.85	36.82 ± 1.49
L3	28.33 ± 7.83	28.79 ± 5.86	38.15 ± 1.69
L4	22.81 ± 5.04	20.17 ± 6.05	31.79 ± 2.07
L5	23.85 ± 7.19	25.64 ± 7.26	33.25 ± 2.17
L6	32.22 ± 7.33	29.14 ± 6.59	41.94 ± 1.94
S1	31.4 ± 9.20	36 ± 8.49	42.57 ± 2.99

## Data Availability

The data presented in this study are available on request from the corresponding author. The data are not publicly available due to privacy restrictions.

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
