# Peer review of "Cystitis Induces Altered CREB Expression Related with Micturition Reflex"

_medicina, 2022, doi:10.3390/medicina58091210_

Round 1

Reviewer 1 Report

- title is confusing, please revised it to be more simpler.
- This is original manuscript; Thus, abstract should be structured including background, method, result and conclusion.
- The bacterial names e.g. Escherichia coli (E. coli) should be specified with italic font throughout the manuscript.
- Figure resolution was low, please upload high quality images with extended legends to fully described the content of this study.
- Please highlighted the novelity of this study.
- The author should discuss about limitation of this study.
- The author should provided an objective conclusion with further perspective to readers for future studies.
- The references were too old, please make revision in discuss using the most relevant references.

Author Response

All authors appreciate you reviewing manuscript. We thoroughly corrected it according to your advice.

- Title is confusing, please revised it to be simpler.

To reflect your opinion, we have clarified the title, ‘Cystitis induces altered CREB expression related with micturition reflex’

- This is original manuscript; thus, abstract should be structured including background, method, result and conclusion.

Thanks for a sharp observation. We indicated and made a division in abstract; ‘Background and objectives’, ‘Materials and methods’, ‘Results’ and ‘Conclusions’.

- The bacterial names e.g. Escherichia coli (E. coli) should be specified with italic font throughout the manuscript.

Thanks you for well-meant advice. We corrected all names to italic text.

- Figure resolution was low, please upload high quality images with extended legends to fully describe the content of this study.

We shared your opinion, and uploaded highest DPI (600) figures. And we gave a full description of each figures. 185th line ~ 202nd line: figures & their legends.

- Please highlighted the novelty of this study.

We agree entirely with your opinion.

Since conceptualization of this article, we recognized that there has been only few studies on the relationship with p-CREB expression and bacterial cystitis for decades. We would like to stress this point in introduction part.

66th line: There is only few study on the relationship between p-CREB expression and cystitis, to the best of our knowledge, this study is the initial report on the p-CREB expression after bacterial cystitis.

- The author should discuss about limitation of this study.

Thanks for the sincere comment. We added the limitations in discussion part.

275th line: There are a few limitations. We couldn’t continue our research to reaffirm consistent results on other neurotrophin or transcription factor of Trk signaling pathways, expected to be closely connected to p-CREB. A few similarly-themed studies on CREB activity in cysti-tis have been reported, but there is still a dearth of research to support the significance of p-CREB. Furthermore, our result might have been underpowered due to relatively small sample size. To overcome this limitations, further large scale, prospective and mul-ti-center trials can verify the precise role of p-CREB alteration which may trigger long-term pathophysiologic changes in micturition reflex.

- The author should provide an objective conclusion with further perspective to readers for future studies.

Thanks for the sincere comment. We added the limitations in discussion part.

283rd line: Based on our results, acute or chronic bacterial cystitis may cause unfavorable changes in micturition reflex, and the novel medicines modulating p-CREB expression would be one solution to care patients complaining of persistent urinary symptom with a history of urinary tract infection.

- The references were too old, please make revision in discuss using the most relevant references.

Thanks for a sharp observation. As we mentioned above, there is only few previous similarly-themed studies for decades. We did our best, and added two recent reference literatures (published in 2013 and 2016).

  1. Kay JC, Xia CM, Liu M, Shen S, Yu SJ, Chung C, et al. Endogenous PI3K/Akt and NMDAR act independently in the regulation of CREB activity in lumbosacral spinal cord in cystitis. Exp Neurol 2013;250:366-75.
  2. Zhang X, Yao J, Gao K, Chi Y, Mitsui T, Ihara T, et al. AMPK Suppresses Connexin43 Expression in the Bladder and Ameliorates Voiding Dysfunction in Cyclophosphamide-induced Mouse Cystitis. Sci Rep 2016;6:19708.

Reviewer 2 Report

Dear Authors,

article brings results of an interesting study on an animal model regarding CREB altered expression due to bacterial cystitis.
Several points should be improved:
1) Language and English style should be improved. There are few  connectors or linking words between a phrase to another, especially in introduction 
2) Methods: if a particular IRB protocol was registered for this article, please add it to the Experimental Animal sub-section
3) Methods: please indicate the software used for statistical analysis (STATA? SPSS? etc..)
4) Results: Table 1 might be improved reporting also the p-value column of inter-group comparisons 
5) Discussion: In discussion there are no strength and limitations section, thus should be implemented

Author Response

All authors appreciate you reviewing manuscript. We thoroughly corrected it according to your advice.

1) Language and English style should be improved. There are few connectors or linking words between a phrase to another, especially in introduction

Actually, this manuscript had been corrected several months before. To reflex your opinion, all authors rechecked the overall grammar and are also interested in the proofreading arranged by MDPI, if necessary.

2) Methods: if a particular IRB protocol was registered for this article, please add it to the Experimental Animal sub-section

In fact, the animal study was proceeded before 2009. Up to that time, it had been legal experiment without the approval of IACUC in South Korea. There has been quite a long interval from study design to paper writing. I received the overall data of the study last year, and noticed that this forgotten study is still worthy enough until now. Authors refined the article, and finally submitted to one of the pioneering articles, 'Medicina'. We obeyed the general regulations of animal studies, as described in the manuscript. After consulting our central laboratory, it is obvious that there is not any evidence of 'approval code or date' about the preceding studies before 2009. To clarify this, additional comments were added in the 'materials and methods'.

81st line: All experimenter do their research legally and followed the protocols approved by the Animal Care Committee of the Animal Center at Kyung Hee University, and complied with the guidelines from the Korean National Health Institute of Health Animal Facility.

3) Methods: please indicate the software used for statistical analysis (STATA? SPSS? etc..)

Thanks for the sincere comment. We added the description of statistical analysis software in materials & methods part.

148th line: The collected data were transformed to meet the requirements of ANOVA and analyzed using SPSS 28.0 software.

4) Results: Table 1 might be improved reporting also the p-value column of inter-group comparisons

Thank you for well-meant advice. All authors thought long and hard about inserting p-value column (e.g. control vs acute cystitis, control vs chronic cystitis and acute vs chronic cystitis).

Finally, we made a decision, and the contents of table 1 and figure 3 are expected to be fairly similar after supplement of p-value column in table 1. Authors intended to increase visibility and comparability of our data through figure 3.

To sum up, it is a pity that we would have to put that on hold.

5) Discussion: In discussion there are no strength and limitations section, thus should be implemented

Thanks for the sincere comment. We added the limitations in discussion part.

275th line: There are a few limitations. We couldn’t continue our research to reaffirm consistent results on other neurotrophin or transcription factor of Trk signaling pathways, expected to be closely connected to p-CREB. A few similarly-themed studies on CREB activity in cysti-tis have been reported, but there is still a dearth of research to support the significance of p-CREB. Furthermore, our result might have been underpowered due to relatively small sample size. To overcome this limitations, further large scale, prospective and mul-ti-center trials can verify the precise role of p-CREB alteration which may trigger long-term pathophysiologic changes in micturition reflex.

Additionally, we added some comments in introduction part to emphasize the novelty of our paper.

66th line: There is only few study on the relationship between p-CREB expression and cystitis, to the best of our knowledge, this study is the initial report on the p-CREB expression after bacterial cystitis.

Round 2

Reviewer 2 Report

Dear Authors,

thank you to have addressed all the previously raised points

Kind regards